# Measuring Children’s Stress via Saliva in Surgical and Endoscopic Procedures and Its Measurement Intention in the Community: Reality-Future Prospects

**DOI:** 10.3390/children10050853

**Published:** 2023-05-10

**Authors:** Maria Grigoropoulou, Achilleas Attilakos, Anestis Charalampopoulos, Smaragdi Fessatou, Efstratios Vamvakas, Anastasia Dimopoulou, Nikolaos Zavras

**Affiliations:** 1Second Health Centre,12132 Peristeri, Greece; 2Department of Pediatrics, Attikon University General Hospital, National and Kapodistrian University of Athens, 10679 Athens, Greece; 3Department of Surgery, Attikon University General Hospital, National and Kapodistrian University of Athens, 10679 Athens, Greece; 4Second Critical Care Department, Attikon University General Hospital, National and Kapodistrian University of Athens, 10679 Athens, Greece; 5Department of Pediatric Surgery, Attikon University General Hospital, National and Kapodistrian University of Athens, 10679 Athens, Greece

**Keywords:** saliva cortisol, planned behavior theory, operated children’s stress

## Abstract

(1) Background: Children who undergo surgical or endoscopic procedures display high levels of stress, and various means are applied to reduce their anxiety. Salivary cortisol (S Cortisol) and salivary alpha-amylase (SAA) are often used as a valid biomarker of stress. The primary purpose of the study was the investigation of stress levels through S Cortisol and S amylase after intervention in surgical or endoscopic procedures (gastroscopy–colonoscopy). The secondary outcomes were the investigation of the intention to adopt new methods of saliva sampling. We collected saliva samples from children subjected to invasive medical procedures, with the aim of applying the Theory of Planned Behavior (TPB) as an intervention means to provide information and education to both parents and children undergoing stressful situations, and assess its efficacy in reducing stress levels. We also aimed at acquiring a better understanding of the acceptability of noninvasive biomarker collection in community settings. (2) Methods: The sample of this prospective study comprised 81 children who underwent surgical or endoscopic procedures at the Attikon General University Hospital, Athens, Greece and 90 parents. The sample was divided into two groups. The first, ‘Group Unexplained’, was not provided any information or education about the procedures, while the second, ‘Group Explained’, was informed and educated based on TPB. Thereafter, 8–10 weeks after intervention, the Theory of Planned Behavior questions were re-completed by the ‘Group Explained’. (3) Results: Significant differences were detected in cortisol and amylase values between the two groups postoperatively after applying the TPB intervention. Saliva cortisol was reduced by 8.09 ng/mL in the ‘Group Explained’ while in the ‘Group Unexplained’ it was reduced by 4.45 ng/mL (*p* < 0.001). Salivary amylase values decreased by 9.69 ng/mL in the ‘Group Explained’ after the intervention phase of the study, while in the ‘Group Unexplained’ they increased by 35.04 ng/mL (*p* < 0.001). The regression explains 40.3% (baseline) and 28.5% (follow-up) of parental intention. The predictive factor of parental intention (baseline) is attitude (*p* < 0.001) and follow-up is behavioral control (*p* < 0.028) and attitude (*p* < 0.001). (4) Conclusions: Providing proper education and information for parents has a positive effect on reducing children’s stress levels. Changing parental attitudes towards saliva collection plays the most important role, since a positive attitude can influence intention and ultimately participation in these procedures.

## 1. Introduction

Pediatric patients are a highly vulnerable population, prone to exhibiting high levels of stress and anxiety when undergoing surgical and endoscopic procedures. It has been shown that between 50% and 75% of children undergoing elective surgery experience perioperative anxiety [1,2,3], which can lead to negative clinical effects [4].

The influencing factors which impact on pediatric patients’ preoperative stress include children’s younger age, previous hospitalizations and the surgical setting itself. However, the factors that play a significant role in high levels of stress among the pediatric population are the anxiety and stress experienced by their own parents [5]. Some of the factors that influence parental stress include their fear of anesthesia, postoperative pain, and lack of knowledge and information regarding the whole procedure [6]. One way to determine the level of stress among pediatric patients is the measurement of cortisol [7]. Progress in biotechnology, in combination with the clinical interest in a user friendlier and non-invasive practice, have led to diagnostic saliva testing [8]. Saliva allows for repeated sampling, is safer in its handling, requires simple equipment and reduces the risk of anemia in children [9]. Moreover, saliva biomarkers can be assessed at home without the presence of medical personnel [10]. Perhaps no other patient population could be more benefited by this progress than the pediatric one [8]. An explanation of this is that saliva is a means of a timely detection of various diseases and conditions in children [9], as demonstrated by several studies, such as the one by Tintor et al. (2023) [10], in which the diagnostic accuracy as a novel biomarker for acute appendicitis of Leucine-rich α-2-glycoprotein 1 (LRG1) in saliva was investigated. The sample of the study constituted of 92 children, 46 of which underwent laparoscopic appendectomy and 46 were without appendicitis, aged between 5 and 17. The results demonstrated a statistically significant increase in LRG1 in the saliva of children with acute appendicitis compared with the control group, allowing for the effective discrimination between acute appendicitis and controls (AUC = 0.85, 95% CI 0.76–0.92, *p* < 0.001) [10].

The hypothalamic–pituitary–adrenal (HPA) axis is responsible for controlling stress through both regulation of synthesis and secretion of cortisol [8]. Elevated cortisol levels are associated with stress symptoms, such as sleep disorders, eating disorders, enuresis [11], depression, isolation, arousal and aggressiveness [12], and salivary cortisol has been determined to be a valid biomarker of stress [13]. Saliva collection is non-invasive and can provide reliable results regarding cortisol levels, given that saliva is highly sensitive to stress due to its protein secretion, which largely depends on the sympathetic nervous system [14].

Although the analysis of saliva has advantages, the collection of the samples must be conducted in a specific way [15]. Cortisol, as a glucocorticoid hormone, activates the hypothalamic–pituitary–adrenal axis when a stressful event is experienced. The production and release of increased cortisol in plasma occurs 15 to 30 min after a stressful event, and 2 min later it appears in saliva. For example, if a child experiences a stressful event at 9:00 a.m., the child’s cortisol response will peak in serum around 9:20 a.m., and in saliva it will do so around 9:22 a.m. Cortisol is secreted in circadian rhythms in 15 to 30 spurts during the day, with the highest levels occurring about 20 to 30 min after morning waking, are reduced to half by noon, and are at the lowest levels by midnight. Changes in cortisol levels are caused by increased physical activity, eating and varying sleep habits, while the intake of certain medications, some diseases and the intake of substances which stimulate the amount of saliva should be taken into account when measuring saliva cortisol. Additionally, changes in cortisol levels are related to age, weight, gender, temperament, and the attitude towards pain. Thus, implementing strategies for reliable saliva sample collection increases the validity of the test results. These strategies include: (1) standardizing the time of sample collection, including baseline samples; (2) using consistent collection materials and methods; (3) controlling for certain drinks, foods, medications, and diagnoses; and (4) establishing procedures and protocols. Other strategies for the laboratory analyses include: (1) selecting the appropriate assay and laboratory; (2) identifying units of measure and norms; and (3) establishing quality controls [16].

However, since children’s stress is highly related to parental stress, addressing the latter through targeted interventions could reduce children’s stress and anxiety [17].

The Theory of Planned Behavior (TPB) constitutes a method which can be used to understand complex health and behavioral phenomena [18]. It targets behavioral change through education and training [19]. Its main aim is to equip individuals with the appropriate information, skills and sense of control to enable them to deal with a stressful situation [20]. 

One of the advantages of TPB is the concept that an attitude is associated with the intention of the individual to adopt a behavioral pattern [21]. Based on this theory, behavioral intention is influenced by three factors: the first is attitude, which is defined as the positive or negative assessment of a behavior; the second concerns subjective norms, which mainly relate to the social pressure felt to exhibit a particular behavior; and finally, perceived behavioral control, which is the ability of the individual to respond [22].

The advantage of this theory as an intervention lies in its potential to predict intention and in its principle, according to which attitudes can change through communication and provision of information, with the added advantage that subjective norms and perceived control can be altered through social support and experience [23]. Based on the aforementioned, the aim of this study was to determine whether the application of TPB as an intervention to parents could ultimately reduce stress levels in children who undergo surgical or endoscopic procedures, as measured through salivary cortisol levels, and whether parents would be willing to allow their children to participate in the diagnostic procedure of saliva collection. Therefore, the primary purpose was the investigation of stress levels through S Cortisol and S amylase after intervention in surgical or endoscopic procedures (gastroscopy–colonoscopy). The secondary outcomes were the investigation of the intention to adopt new methods of saliva sampling.

The main hypothesis of the study was that parental intention to permit their children’s participation in new diagnostic procedures is influenced by their attitudes towards such procedures. Adequate training of parents would lead to greater intention, with the added benefit of reduced perioperative stress in pediatric populations.

## 2. Materials and Methods

A prospective study was conducted in children who underwent surgical or endoscopic (gastroscopy–colonoscopy) procedures and their parents in the pediatric surgical clinic of Attikon General University Hospital in Athens, Greece from January 2020 to January 2022.

The adequate number of the study participants was determined using G* POWER 3.1.9.7 as follows: A priori power analysis and statistical test (means: difference between two independent means—two groups), tail(s) = 2, effect size Cohen’s d = 0.65, an err prob = 0.05, power (1 − β err prob) = 0.80, allocation ratio N2/N1 = 1, sample size group 1 = 39, sample size group 2 = 39. Based on these assumptions, the minimally needed total size was calculated at 78 subjects in total.

During the design of the study, emphasis was given to the fact that the laboratory analyzed 90 samples each time, a number corresponding to 3 samples T1, T2, and T3 from each child with a maximum number of 30 children, based on which the SPSS database was structured. The distribution was as follows: in the 90 cells of SPSS, 45 cells (value = 0, labels = ‘Group Unexplained’) and 45 cells (value = 1, labels = ‘Group Explained’) were randomly assigned. After SPSS was distributed, folders with a serial number according to SPSS were created. When the research began, each child’s admission to the hospital corresponded to a specific file. The order was strictly followed. Each folder contained the questionnaire (TPB) distributed to the parents, three SaliCap Tubes for the child’s saliva samples, and the child’s inclusion group, of which the parents were not aware.

Children’s surgical stress is affected by the type of operation as demonstrated by the research of Jukić et al. (2019) [24]. For this reason, during the design of the study, it was decided that the study sample would be children undergoing simple routine surgeries or endoscopic procedures (gastroscopy–colonoscopy) so as to have similar stress levels.

### 2.1. Inclusion and Exlusion Criteria

Age between ≥4 years and ≤15 years and, for both parents and children, fluent understanding and speaking the Greek language were set as inclusion criteria. Exclusion criteria were determined as follows: diagnosis of attention deficit hyperactivity disorders, autism, Down syndrome, and need for dental care, treatment strategies based on pharmacological therapies, as well as the use of steroids orally or intravenously for various diseases.

### 2.2. Measures

#### 2.2.1. Salivary Parameters

Salivary cortisol and S alpha-amylase levels were measured at T1, after T2 and T3. The saliva samples were collected using SaliCap (also referred to as passive drool) and frozen at −80° until analysis in the laboratory.

#### 2.2.2. Comments on Elisa Method

Standard saliva cortisol ELISA (0.1–30.0 ng/mL range) kits were purchased from LDN (Nordhorn, Germany). The free Cortisol in Saliva ELISA Kit is a solid phase enzyme-linked immunosorbent assay (ELISA) based on the principle of competitive binding. The microtiter wells are coated with a polyclonal rabbit antibody directed against the cortisol molecule. ELISA was carried out according to the manufacturer’s instructions. Briefly, the samples (50 μL) are dispensed in the coated wells and incubated with the enzyme conjugate (cortisol conjugated to horseradish peroxidase, 50 μL). During incubation (60 min, at room temperature on a plate shaker, 900 rpm), endogenous cortisol of a patient sample competes with the enzyme conjugate for binding to the coated antibody. The unbound conjugate is removed by washing the wells. Subsequently, the substrate solution (200 μL, TMB) is added, and the color development is stopped (50 μL, 2N HCl) after a defined time (30 min). The intensity of the color formed is inversely proportional to the concentration of cortisol in the sample. The absorbance is measured at 450 nm with a microtiter plate reader (Biochrome ASYS Expert 96, Vienna, Austria). The results for the samples are determined using the standard curve and 4-parameter logistics calculation. According to the manufacturer’s data on performance characteristics for saliva cortisol ELISA kit, the analytical sensitivity is 0.019 ng/mL, the intra- and inter-assay coefficient of variation (CV) ranges from 4.1 to 7.1% and 4.2 to 9.1%, respectively.

Standard alpha-amylase ELISA (12–500 U/mL range) kits were purchased from LDN (Germany). The ELISA test kit provides a quantitative in vitro assay for free alpha amylase in human saliva. The test kit contains microtiter strips each coated with anti-rabbit antibodies. ELISA was carried out according to the manufacturer’s instructions. Briefly, in the first reaction step, diluted patient samples (20 μL) are pipetted into the reagent wells together with peroxidase-labelled alpha-amylase (100 μL) and a specific rabbit anti-alpha amylase antibody (100 μL). During incubation (60 min, at room temperature on a plate shaker, 400 rpm), alpha amylase from the patient sample and the labelled alpha amylase in the conjugate compete for the free binding sites of the specific antibody. In the third incubation step, the bound peroxidase catalyzes a color reaction with the peroxidase substrate tetramethyl benzidine (TMB, 100 μL). The intensity of the color formed is inversely proportional to the concentration of alpha amylase in the sample. The absorbance is measured at 450 nm with reference filter at 620 nm, with a microtiter plate reader (Biochrome ASYS Expert 96, Austria). The results for the samples are determined using a standard curve and 4-parameter logistics calculation. According to manufacturer’s data on performance characteristics for alpha-amylase ELISA kit, the analytical sensitivity is 3.6 U/mL, the intra- and inter-assay coefficient of variation (CV) ranges from 3.6 to 5.5% and 4.2 to 9.6%, respectively.

### 2.3. The Theory of Planned Behavior (TPB)

The first form used, consisting of 14 items, regarded the demographic data of the parents, including questions regarding their sex, age, nationality, marital status, educational level, occupation, place of residence and annual family income as well as the number of their children, and the type of surgery their children would undergo.

The second part regarded the Theory of Planned Behaviour developed by Ajzen and Fishbein [25,26,27], which is free for research use, and was adjusted for the needs of the present study. The construction of the questionnaire, so as to measure the variables in the TPB model, includes nine phases according the manual of Francis et al. (2004) [28]. Questionnaire reviews [28,29,30,31,32,33] were included to construct the questionnaire. The final version of the survey contained minor revisions to the wording and order of the questions based on similar Greek questionnaires administered to children’s parents [34], recommendations from professionals, and the findings from the pilot study.

The second draft of the questionnaire was submitted to 10 parents with characteristics similar to those of the target population. The parents completed the questionnaire twice (with an interval of at least two weeks) so that both its stability (testing–retesting) and the internal consistency (Cronbach’s alpha) of the questions of the multiple sections could be determined. The reliability of testing and retesting using the parametric Pearson’s correlation test for the study’s variables were intention (r = 0.949, *p* < 0.001), attitudes (r = 0.994, *p* < 0.001), perceived control (r = 0.972, *p* < 0.001) and subjective norms (r = 0.902, *p* < 0.001). Internal consistency was verified using the Cronbach’s alpha coefficient with the following values: intention a = 0.886; attitudes a = 0.987; perceived control a = 0.842; and subjective norms a = 0.916. The final version of the tool was named Questionnaire Regarding Parental Acceptance of Receiving their Children’s Saliva Biomarkers. It constitutes 18 items which are divided into the following subscales.

Intention: It measures parental intention towards saliva biomarkers collection with 3 items of a 7-point Likert scale, where 1 is unlikely and 7 is likely. Higher mean scores indicate higher parental intention.

Attitude: This subscale concerns parental attitudes towards saliva biomarkers collection with 8 items, 4 of which focus on saliva biomarkers and the remaining 4 focus on whether saliva sampling affects general health. All items are of a 7-point Likert scale. More specifically, this subscale measures their emotions regarding their children with provided ranging from ‘1 = unimportant’ to ‘7 = important’, ‘1 = unpleasant/stressful’ to ‘7 = pleasant’, ‘1 = bad’ to ‘7 = good’, ‘1 = irresponsible’ to ‘7 responsible’, ‘1 = harmful’ to ‘7 = beneficial’, ‘1 = worthless’ to ‘7 = worthy’, ‘1 = unnecessary’ to ‘7 = necessary’ and ‘1 = insignificant’ to ‘7 = significant’. Higher mean scores indicate a more positive parental attitude.

Subjective norms: 4 items are included in this subscale regarding parental perception of subjective norms which regard agreement or disagreement with the advice and prescriptive rules about saliva biomarker collection. It evaluates the subjective norms which are associated with adult intention to allow saliva biomarker collection from their children. The items are in a 7-point Likert scale, with 1 indicating total disagreement and 7 indicating total agreement. Higher mean scores indicate stronger subjective norms.

Perceived behavioural control: 3 items assess parental perception regarding their ability to control saliva biomarker collection. More specifically, they evaluate perceived behavioral control, such as self-efficacy assessing the extent that adults perceive themselves as capable of showing a behaviour, through a 7-point Likert scale of the degree of ease they felt regarding their children’s participation in saliva biomarker collection. In this subscale, the provided answers range among the following: ‘1= extremely difficult’ to ‘7 = ‘extremely easy’, ‘1 = extremely uncertain’ to ‘7 = extremely certain’, ‘1= extremely low sense of control’ to ‘7 = extremely high sense of control’. Higher mean scores indicate higher levels of perceived behavioural control.

One item regards the future subjective perception for future behaviour (1 = unlikely to 7 = likely).

All internal consistency coefficients are >0.6 and are acceptable according to the main analysis of constructing questionnaires based on the theory of planned behaviour: A manual for health service researchers [28]. Qualitative descriptors used for values/ranges of values of Cronbach’s alpha were reported in papers according to Taber (2018) [35] in leading science education journals. Thus, alpha values were described as excellent (0.93–0.94), strong (0.91–0.93), reliable (0.84–0.90), robust (0.81), fairly high (0.76–0.95), high (0.73–0.95), good (0.71–0.91), relatively high (0.70–0.77), slightly low (0.68), reasonable (0.67–0.87), adequate (0.64–0.85), moderate (0.61–0.65), satisfactory (0.58–0.97), acceptable (0.45–0.98), sufficient (0.45–0.96), not satisfactory (0.4–0.55) and low (0.11) [35].

Internal consistency was assessed for each of the TPBs at both Time 1 and Time 2 using Cronbach’s alpha (α) developed by Lee Cronbach in 1951. It is expressed as a number between 0 and 1, and is widely accepted for assessing the internal consistency of questionnaires since all questions in a test measure the same concept. Simply put, the construction of the questions is correlated within the test, and reliability estimates indicate the amount of measurement error, that is, the correlation of the test with itself. Squaring this correlation and subtracting from 1.00 produces the index of measurement error. For example, if a test has a reliability of 0.80, there is 0.36 error variance (random error) in the scores (0.80 × 0.80 = 0.64; 1.00 − 0.64 = 0.36 [36]. In the hospital, the reliability levels of each subscale were as follows: 0.97 for attitude, 0.79 for subjective norms, 0.87 for perceived behavioral control and 0.92 for intention. The range of reliability (0.79–0.97) indicates high internal consistency.

In the community, the reliability levels of each subscale were as follows: 0.93 for attitude, 0.72 for subjective norms, 0.89 for perceived behavioral control and 0.94 for intention.

### 2.4. Sample, Data Collection and Intervention Procedure

The sample of the study involved ninety children (4–15 years) and ninety parents (one of the two parents of each child). Three saliva samples were collected during the following hours: (T1): 7:00 a.m.–9:00 a.m. in the clinic room, (T2): 9:00 a.m.–12:00 p.m. just before anesthesia administration, and (T3): 5:00 p.m.–7:00 p.m. The invasive procedures usually finished between 2:00 p.m. and 4:00 p.m. The third saliva collection took place between 5: p.m. and 7:00 p.m. so as to ensure that the collection was performed at the same period from all the children and to have the margin to implement the intervention in the ‘Group Explained’. Saliva samples were collected using SaliCap, spitting whole saliva—a method also referred to as passive saliva. The children, following the researcher’s guidance, spat into SaliCap Tubes, the capacity of which is 1.5 mL (quantity sufficient for analysis according to the laboratory’s instructions). To ensure the validity of the results, 1.5 cc of saliva was obtained using the passive saliva technique. To ensure the reliability of the measurements, the baseline levels for the child were measured with the first sampling time (T1) together with the response of the child to the stressful event; the difference before intervention T2 (mean difference to S Cortisol and S alpha-amylase T2–T1) and after the operation T3 (mean difference of S Cortisol and S alpha-amylase level T3–T2) were measured, which also measured whether the intervention affected the levels. With regard to the TPB questionnaire, all parents were provided with instructions, after which they completed the questionnaire before the collection of the first saliva sample. The non-pharmacological intervention was allocated to Risk Factors and Prediction of Preoperative Anxiety in Children in order to understand the risk factors for children and parents who are likely to develop significant preoperative anxiety. These risk factors fall into three categories, the first being Child Factors, which include, besides the child’s age, their temperament, their previous medical encounters, as well as their biologically based vulnerabilities and the quality of the relationship between themselves and their parents. The other two categories are Parental Factors and Perioperative Environmental Factors. Preoperative interview provides the healthcare professional with the opportunity to assess the child, develop a relationship with them and their family, provide them with a detailed plan of the operation and reassurance that their child will be well protected [37].

The questions of TBP followed the Information Provision Model of Jaaniste et al. (2007) which concerns the processes involved in the provision of information with the purpose of providing the needed preparation prior a medical procedure [38] (Table 1).

The subjects were then randomly divided into two groups.

The Study Groups

The first, ‘Group Unexplained’, included 45 children and one of their parents: a hardcopy of the questionnaire (TPB) was provided to them to fill in. Then, three saliva samples were collected from the child during the following hours: (T1): 7:00 a.m.–9:00 a.m. in the clinic room, (T2): 9:00 a.m.–12:00 p.m. just before anesthesia administration, and (T3): 5:00 p.m.–7:00 p.m. This group received no explanation related to the surgical or endoscopic procedure. No pre-intra and postoperative information was provided.The second, ‘Group Explained’, also comprised 45 children and one of their parents: a hardcopy of the questionnaire (TPB) was provided to them to fill in. After the completion of the questionnaire (TPB), the researcher stayed with the parents so as to provide an explanation for these particular issues, because the understanding of possible hesitations or fears of the participants regarding an operation and the use of biological samples is important. The Theory of Planned Behavior (TPB) in research allows for a complete and in-depth understanding of the multifaceted perceptions regarding the understanding of complex phenomena. It explores the influence of internal factors (such as parents’ beliefs), external factors (such as barriers from the hospital environment itself) and the influence of ‘significant others’ (such as relatives, friends or health professionals) on parents’ intention to adopt a participative behavior. Through dialogue, questions were answered, and the thoughts, needs, expectations and experiences of both the child and parent were recognized. This process also empowers both the parent and the child to gain confidence and a sense of control over a stressful situation for a smooth outcome. At the same time, information was obtained regarding the acceptance of the collection for biological samples from saliva.

The main demographical data for both groups are presented in Table 2.

Parental involvement in the research provides parents with active roles during their child’s hospitalization. In this way, their own stress, and subsequently their child’s stress, is reduced, since parental behavior is inextricably linked to their child’s behavior.

Some days prior the invasive procedure, basic medical information was provided to both groups by the doctor. The doctor was not aware of the group to which the child and the parent belonged in order for them not to be influenced in the case they knew that the child and the parent belonged to the first ‘Group Unexplained’. The parents during their child’s admission to the hospital confirmed that they wanted to participate in the research. The parents in the first ‘Group Unexplained’ were informed that questions were allowed to be asked only of the hospital staff (and not the person responsible for the collection of both questionnaires and saliva). This was implemented in order for the children’s levels of salivary cortisol and s-alpha-amylase not to be affected.

During the second and the third sampling, three children in the ‘Group Unexplained’ and six children in the ‘Group Explained’ were unable to co-operate, and were therefore excluded from the study. As a result, the final sample size in the ‘Group Explained’ was 39 children and in the ‘Group Unexplained’ it was 42 children. Their parents, however, were not excluded for the purpose of assessing parental intention to allow their children’s participation in new diagnostic methods such as saliva collection. Thereafter, 8–10 weeks after intervention, the Theory of Planned Behavior questions were re-completed by the first ‘Group Explained’ (two questionnaires were not completed).

### 2.5. Statistical Analysis

The data were analyzed using the statistical software IBM SPSS, version 22, of IBM Corp, New York, NY, USA. Constant variables are presented as means and standard deviations (SD). Categorical variables are presented as frequencies and percentages. To determine whether the variables were normally distributed, the Kolmogorov–Smirnov test was used. For the statistical analysis, *t*-test, Pearson’s r, and Multiple Linear Regression were applied. The statistical significance was set at *p* < 0.05.

## 3. Results

In the first analysis of this study, the salivary cortisol level (SCL) and salivary alpha-amylase (sAA) were measured in children who underwent surgical or endoscopic procedures and their effects on the procedure were evaluated. Mean difference of salivary cortisol and salivary alpha-amylase reactivity was normally distributed (Kolmogorov–Smirnov, *p* > 0.05).

Eighty-one children (52 boys and 29 girls) were included into the study. Children were divided into two groups (‘Group Unexplained’ and ‘Group Explained’). Table 3 presents the sociodemographic data of the children and the influence of the intervention on salivary cortisol and s alpha-amylase levels before the intervention. There was no statistically significant difference between the groups, demonstrating homogeneity in the studied sample.

The comparisons of the two groups (Table 3) were performed with Student’s *t*-test. There was no significant difference between the values of initial stress in the control group versus the experimental group. On the contrary, after implementing the intervention, the comparison of salivary cortisol and amylase levels between the two groups showed that in the ‘Group Explained’, salivary cortisol values decreased by 8.09 ng/mL after intervention phase of the study, while in the ‘Group Unexplained’, they decreased by 4.45 ng/mL. This finding is of a statistical significance (t = 3.395, df = 79, *p* = 0.01; effect size 0.73). Concerning the salivary amylase values, they decreased by 9.69 ng/mL after intervention phase of the study, while in the ‘Group Unexplained’, they increased by 35.04 ng/mL (t = 3.32, df = 79, *p* = 0.01; effect size 0.67) (Table 3 and Figure 1).

In the second analysis, exploring results of data analysis included two multiple regression tests. Skewness/Kurtosis coefficients and Kolmogorov–Smirnov indicate a multivariate normality of the data.

Correlation analysis shows which independent variables—prognostic factor (attitudes, control and subjective norms)—directly influence the dependent variable—outcome (intention). More specifically, baseline intention correlated positively with attitude (r = 0.0618, *p* < 0.001), perceived behavioral control (r = 0.288, *p* < 0.006) and subjective norms (r = 0.257, *p* = 0.014), and follow-up intention correlated positively with attitude (r = 0.326, *p* < 0.033), perceived behavioral control (r = 0.372, *p* < 0.014) and subjective norms (r = 0.305, *p* = 0.047).

In general, the sampled parents showed a positive intention towards the acceptability of noninvasive biomarker collection in the hospital (mean = 5.01) and the community (mean = 5.29) (Table 4).

Regarding the intention of parents to allow their children to participate in new diagnostic procedures, particularly saliva collection, to measure stress levels based on TPB as shown in Table 5, it was observed that the regression model explained 40.3% of parental intention. This suggests that if attitudes, subjective perception of control, and subjective behavioral norms are known, we can predict 40.3% of parental intention to participate in saliva collection. Moreover, the predictive factor of parental intention is attitude (t = 6.675, *p* < 0.001, Beta = 0.737), but not perceived behavioral control nor subjective norms, which can affect parental intention only indirectly.

In the community, it was observed that the regression model explained 28.5% of parental intention. The predictive factor of parental intention is behavioral control (t = 2.28, *p* < 0.028, Beta = 0.319) and attitude (t = 2.33, *p* < 0.001, Beta = 0.316) (Table 5).

## 4. Discussion

Pediatric patients have high stress levels prior to surgical or endoscopic procedures, as measured by salivary cortisol, which was determined to be a well-accepted diagnostic procedure. Saliva analysis is of increasing interest in clinical diagnosis and disease prognosis, especially in the pediatric population [39,40,41,42]. Previous research has shown that since stress through the central nervous system increases the levels of biomarkers, salivary cortisol can effectively be used to assess the stress levels [43].

One way to reduce the stress levels experienced by pediatric populations is the application of TPB. In the present study, TPB was shown to have a positive influence on stress reduction among children who underwent surgical or endoscopic procedures. More specifically, when parents were provided with all the necessary information about the procedure and were educated on how to support their children, children’s salivary cortisol and salivary alpha-amylase levels were significantly lower after the procedures compared with those of children whose parents did not participate in the intervention (*p* = 0.001). This could be attributed to the fact that the application of a method that relies on providing information and education regarding a medical procedure helps reduce the fear and anxiety experienced by children [21,44].

The greater cognitive capacity of older children before intervention resulted in S Cortisol level T2–T1 being decreased by 5.21 (*p* = 0.013). In our study, older children’s pre-intervention cognitive ability was observed and specifically older children’s T2–T1 cortisol levels decreased by 5.21 compared to younger children aged <11 years, in whom the levels decreased by 2.44 (*p* = 0.013). This is explained by the fact that younger children lag behind in stress-reducing coping strategies and rely more on comforting support from adults. It appears that children’s fears are a function of social adaptability and cognitive development in this situation. In other words, older children are more capable of adopting and applying various stress coping strategies. There is verbal interaction between the older children and the health professional, but also a will to learn. For example, they want an explanation for the existence of higher levels of perioperative stress. Moreover, it is important to predict whether a child is capable of responding emotionally to a stressful situation based on their temperament. Thus, mild-mannered or shy children appear to be more anxious in stressful situations, as shown by adrenocortical stimulation [45]. These findings are in line with other research studies, which support that this method can be used to understand complex health and behavioral phenomena aiming at behavioral change. The way to achieve this is through equipping individuals with appropriate skills and information to enable them to successfully deal with a stressful situation [18,19,20,21,22,23].

In addition, TPB as an intervention to reduce stress in the pediatric population is based on the fact that children are affected by their parents’ stress. When parental stress is reduced by such an intervention, their children also experience less stress [1,6,46]. The preparation of both children and parents before a medical procedure, according to their needs, can improve their general health and reduce their stress. Although several approaches are effective, the dialogue before surgery helps children acquire a good psychological state [12]. In a study by Wennstöm et al. (2011) [41], 93 children between 5 and 11 years of age undergoing surgery (79 boys and 14 girls) were randomly divided into three groups: (i) usual perioperative care (*n* = 31), (ii) usual perioperative care with information provision (n = 31), and (iii) perioperative dialogue (*n* = 31). Saliva was sampled for cortisol analysis preoperatively and perioperatively. It was observed that low cortisol values postoperatively correlated with preoperative talk (*p* = 0.003) [41].

In a study by Volkan et al. (2019) [42], 184 children aged between 8 and 18 years who underwent gastroscopy had the procedure explained to them. They were subsequently divided into two groups. Only one short explanation of the procedure was provided to the ‘Group Unexplained’, while more detailed procedural information was provided to the ‘Group Explained’. Saliva sampling was conducted both before and after endoscopy. Cortisol levels before and after endoscopy in the ‘Group Unexplained’ were higher (*p* < 0.01 and *p* = 0.02, respectively) in relation to the ‘Group Explained’. A positive correlation was detected between saliva cortisol levels before endoscopy and stress (r = 0.360, *p* < 0.01) [42]. Similar results were observed in the research of Kara et al. [40], in which 119 children (10.9 ± 3.2 years, 43.7% boys) who underwent endoscopy and 85 healthy children (11.8 ± 2.8 years, 45.1% boys) were included. Saliva was collected at specific time points to measure stress levels: (a) between 08:00 and 13:00 a.m., before endoscopy; (b) 30 min before endoscopy; and (c) 2 h after endoscopy. Pre-endoscopy salivary cortisol levels were significantly higher than normal levels even after endoscopy (*p* < 0.001 for each) [40].

Parental intention in the hospital to allow their children to participate in new diagnostic tests in the present study was associated with their attitude towards such procedures. Parental attitude was determined to be the most significant predictive factor of parental intention (t = 6.675, *p* < 0.001, Beta = 0.737), while behavioral control and subjective norms can affect parental intention only indirectly. Parental intention in the community was associated with their behavioral control (t = 2.28, *p* < 0.028, Beta = 0.319) and attitude (t = 2.33, *p* < 0.001, Beta = 0.316). Perceived behavioral control reflects a person’s attitude. Factors such as the cost, flexibility of hours, location, and transportation may not be under the control of individuals [47].

Condon’s review of 31 articles highlighted the lack of available information regarding attitudes, norms and perceived control beliefs, which influence the acceptance and feasibility of saliva collection in children. Identifying misconceptions regarding the collection of saliva for cortisol analysis, such as children’s safety or inappropriate use of the sample, is a study area for researchers that could help to inform participants, eliminate any insecurities, and ultimately gain consent to the procedures [18]. Furthermore, through appropriate interventions which focus on the communication with children and their families, trust towards healthcare professionals is enhanced. resulting in their becoming used to the hospital and surgical environment [48].

The results of this study prove exactly that, and provide opportunities for developments in clinical practice so as to better address the preoperative distress of children. Moreover, the significance of this prospective study lies in the determination of the factors which influence parental willingness to allow their children to provide biospecimens, as well as in the significance of the use of TPB to appropriately manage children’s psychological distress amidst stressful situations. Lastly, the measuring of stress levels through saliva cortisol in children to test whether the application of a TPB intervention can reduce the psychological distress of children preoperatively is conducted for the first time; thus, it offers a new perspective on how to approach this sensitive pediatric population and their families.

## 5. Study Limitations

We acknowledge that the present study has some potential limitations, such as reduced co-operation in the sampling methodology at younger ages. Thus, ages up to three years were excluded. The period in which the study was conducted coincided with the COVID-19 pandemic, during which, on the one hand, surgeries and endoscopies were not performed at the hospital for some periods of time, and on the other hand, parents were afraid to present themselves and their children to the hospital, which explains the limited number of participants. Another limitation is the lack of choice between laboratories for saliva analysis since there is a relatively limited number of private laboratories equipped to analyze cortisol samples, which also leads to increased cost.

## 6. Conclusions

In conclusion, analysis of cortisol concentrations in saliva is an accurate method for assessing the stress levels of pediatric patients and it helps to better understand children’s psychology, thus preventing potential complications during a surgical or endoscopic procedure.

The results of this study provide evidence for the effectiveness of TPB as an intervention in reducing salivary cortisol levels. The proper TPB-based education of parents along with the provision of information has a positive effect on reducing children’s stress levels. Changing parental attitudes towards saliva collection plays the most important role and influences intention and, ultimately, participation in such procedures.

Finally, the results support the feasibility of a population-based approach towards health promotion through mentoring programmes. Saliva as a means to detect biomarkers contributes to the progression of healthcare. However, its analysis is demanding. In the future, point-of-care diagnostics will provide healthcare professionals with the possibility to obtain valid and fast results at the point of patient care. This research enriches knowledge about accepting saliva biomarkers and also indicates the need for more advanced technologies in the future.

## Figures and Tables

**Figure 1 children-10-00853-f001:**
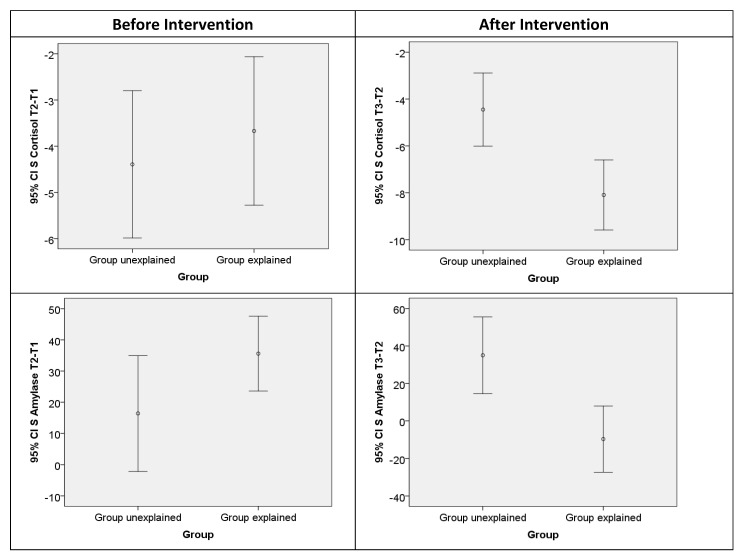
Comparison of S Cortisol and Amylase score distributions between the study groups.

**Table 1 children-10-00853-t001:** Dialogue interview with the most common questions.

Questions	Explanation
Which are their previous medical experiences?	Both children and parents may develop inaccurate images/scenario. Therefore, correct information was provided to improve, process, expand and correct the existing scenarios.
What expectation do children have during a medical procedure?	Correct information was provided to minimize the discrepancy between expectation and reality.
Whether salivary biomarkers testing is important, beneficial, pleasant and necessary.	Both children and parents were provided with information that the procedure is painless with many advantages so as to improve their attitude towards this particular method. Since parents influence their children, it is important that they adopt positive attitudes of managing a situation through the provision of information, communication or persuasive techniques.
How easy do you think the test is? How confident are you in the case you decide to have your child tested for saliva biomarkers? How much do you feel it is up to you to have your child tested for saliva biomarkers? Will you manage to have your child tested?	Since perceived behavioral control is important, parents were properly trained in saliva sampling. The more individuals control the factors in order to perform a particular behavior, the more likely they are to engage in such behavior helping their children engage themselves in these behaviors.
Which is their sense of control and how they cope?	Through the provided information, their sense of control and their solving of focused problems were enhanced.
Further information (explicit or implicit) was collected, e.g., the parent–child relationship. It was also important to identify whether friends and relatives would live to have their child tested for salivary biomarkers, and whether health professionals’ advice is extremely important to parents.	Emphasizing proactive behaviors by influential people leads to an effective change towards a positive direction. Subjective norms had an indirect correlation with parents’ intentions for their children’s behaviors.
Questions about satisfaction or non-satisfaction regarding the number of people in the hospital room, and the waiting time between admission and the start of the procedure were asked. That was because the environment affects the psychology of both children and parents.	The explanations which were provided and regarded the fact that they would be in hospital for a short period of time and that their primary goal is to get well reassured them.
Do you intend to have your child tested for biological saliva markers?	Parental intention was predicted by attitude, subjective norm, and perceived control, and the explanations provided were consistent with previous ones.

**Table 2 children-10-00853-t002:** Characteristics of the sample.

		Group Unexplained *n* = 45	Group Explained *n* = 45
Parent’s gender	Male	15	13
	Female	30	32
Parental education	High School Degree or Less	18	19
	College Degree or More	27	26
Parental income	<10.000	18	15
	>10.000	27	30
Child’s gender	Male	30	28
	Female	15	17
Child’s age	<11	16	21
	>12	29	24
Intervention	Endoscopy	28	31
	Surgical (including):	17	14
	splenic cysts	1	
	phimosis	2	2
	cryptorchidism	3	3
	umbilical hernia	1	
	inguinal hernias	4	
	cysts of the testis		1
	ear reconstruction		1
	liver biopsy	1	2
	varicoceles	1	
	pilonidal cyst	1	2
	appendicitis	1	
	excision of minor lumps	1	
	esophageal achalasia		1
	hydrocele		1
	testicular torsion		1
	biliary	1	

**Table 3 children-10-00853-t003:** Characteristics of the cohort. The effect of external variables on salivary cortisol and salivary alpha-amylase levels.

									Before Intervention				After Intervention			
		n	S Cortisol			S A Amylase			Mean Difference of S Cortisol Level Τ2–Τ1		Mean Difference of S A Amylase Level T2–T1		Mean Difference of S Cortisol Level Τ3–Τ2		Mean Difference of S A Amylase Level Τ3–Τ2	
			T1	T2	T3	T1	T2	T3								
			Mean (SD)	Mean (SD)	Mean (SD)	Mean (SD)	Mean (SD)	Mean (SD)	Mean (SD)	*p* Value	Mean (SD)	*p* Value	Mean (SD)	*p* Value	Mean (SD)	*p* Value
Intervention	Surgical	29	14.17 (4.68)	11.38 (4.14)	5.27 (5.50)	66.00 (56.96)	102.40 (72.58)	105.10 (82.71)	−2.79 (4.02)	0.94	36.40 (51.48)	0.154	−6.11 (6.14)	0.905	2.70 (57.60)	0.261
	Endoscopy	52	14.07 (5.36)	9.32 (4.92)	3.07 (3.48)	76.64 (65.20)	96.27 (60.38)	115.80 (81.27)	−4.74 (5.41)		19.64 (49.65)		−6.26 (4.54)		19.53 (67.47)	
Parental education	High School Degree or Less	32	14.68 (5.16)	9.99 (4.79)	4.25 (4.79)	70.00 (52.16)	90.46 (54.8)	103.69(66.03)	−4.69 (5.03)	0.351	20.46 (45.78)	0.460	−5.17 (5.17)	0.518	13.35 (62.88)	0.986
	College Degree or More	49	13.73 (5.08)	10.10 (4.75)	3.60 (4.17)	74.67 (4.17)	103.69 (70.3)	117.30 (90.36)	−3.62 (5.02)		29.02 (53.77)		−6.50 (5.14)		13.60 (65.79)	
Parental income	<10.000	30	13.54 (5.04)	10.45 (5.09)	5.34 (5.70)	68.40 (52.37)	92.47 (53.88)	105.52 (70.69)	−3.09 (4.57)	0.190	24.07 (54.71)	0.832	−5.11 (4.83)	0.143	13.05 (63.74)	0.962
	>10.000	51	14.43 (5.15)	9.83 (4.55)	2.98 (3.18)	75.43 (67.71)	101.99 (70.46)	115.76 (87.60)	−4.61 (5.23)		26.56 (48.62)		−6.85 (5.24		13.77 (65.19)	
Child’s gender	Male	52	13.78 (5.25)	10.07 (4.72)	3.87 (4.48)	71.07 (57.13)	99.20 (63.88)	109.09 (81.83)	−3.71 (4.77)	0.431	28.12 (44.86)	0.557	−6.20 (4.90)	0.991	9.89 (60.39)	0.502
		29	14.68 (4.85)	10.05 (4.89)	3.83 (4.35)	75.98 (71.43)	97.16 (67.07	117.14 (81.91)	−4.64 (5.48)		21.18 (60.25)		−6.21 (5.62)		19.98 (71.33)	
Child’s age	<11	34	12.74 (5.11)	10.30 (5.34)	3.86 (5.18)	78.19 (59.13)	114.67 (60.42)	120.45 (80.18)	−2.44 (4.60)	0.013	36.48 (54.78)	0.102	−6.43 (6.05)	0.738	5.79 (69.31)	0.361
	>12	47	15.09 (4.91)	9.89 (4.30)	3.85 (3.82)	68.95 (64.72)	86.75 (65.62)	105.83 (82.64)	−5.21 (5.03)		17.80 (46.48)		−6.04 (4.42)		19.08 (60.49)	
Group	Unexplained	42	14.78 (4.91)	10.39 (4.62)	5.94 (5.27)	93.70 (74.82)	110.09 (74.42)	145.12(89.20)	−4.39 (5.12)	0.521	16.39 (59.56)	0.084	−4.50 (5.01)	0.001	35.04 (65.80)	0.001
	Explained	39	13.38 (5.26)	9.71 (4.89)	1.61 (0.98)	50.35 (33.33)	85.95 (50.09)	76.26 (53.42)	−3.67 (4.95)		35.60 (37.09)		−8.09 (4.62)		−9.69 (54.37)	

Statistical significance, T1 = 1st sampling, T2 = second sampling, T3 = 3rd sampling, SD, standard deviation.

**Table 4 children-10-00853-t004:** Descriptive statistics of data.

							Skewness		Kurtosis		Kolmogorov–Smirnov ^a^
	Construct	*n*	Min	Max	Mean	Std.	Statistic	Std. Error	Statistic	Std. Error	
Baseline Hospital	intention	90	2.00	7.00	5.01	1.16	−0.37	0.25	−0.29	0.50	0.200
	attitudes	90	4.00	7.00	5.62	0.80	−0.19	0.25	−0.80	0.50	0.200
	control	90	2.67	6.33	4.68	0.96	−0.20	0.25	−0.62	0.50	0.200
	subjective norms	90	3.75	6.50	5.05	0.61	0.01	0.25	−0.30	0.50	0.099
Follow-up community	intention	43	3.33	7.00	5.29	0.92	0.02	0.36	−0.33	0.71	0.053
	attitudes	43	4.75	7.00	5.96	0.58	−0.22	0.36	−0.36	0.71	0.051
	control	43	3.67	6.00	5.07	0.63	−0.23	0.36	−0.64	0.71	0.064
	subjective norms	43	3.50	6.00	4.90	0.62	−0.31	0.36	−0.46	0.71	0.090

^a^. Lilliefors Significance Correction.

**Table 5 children-10-00853-t005:** Regression analysis TPB hospital (baseline) and community (follow-up).

Correlation Coefficient (R)		Determination Coefficient (R Square)		Adjusted Determination Coefficient (Adjusted R Square)						
A	B	A	B	A	B					
0.635	0.534	0.403	0.285	0.382	0.230					
ANOVA ^a^										
	Sum of Squares		Df		Mean Square		F		P	
	A	B	A	B	A	B	A	B	A	B
Regression	47.85	10.23	3	3	15.95	3.41	19.34	5.178	<0.001 ^b^	0.004 ^b^
Residual	70.91	25.68	86	39	0.83	0.66				
Total	118.77	35.91	89	42						
	Beta		t		P					
	A	B	A	B	A	B				
Attitudes	0.74	0.316	6.68	2.33	<0.001 *	0.025 *				
Control	−0.20	0.319	−1.6	2.28	0.112	0.028 *				
Subjective norms	0.01	0.206	0.13	1.47	0.897	0.151				

A = hospital, B = community, a = dependent variable: intention, b = predictors: (constant),attitudes, control and subjective norms, * statistical significance.

## Data Availability

Data sharing is not applicable to this article due to privacy and ethical restrictions.

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
