# Peer review of "Measuring Children’s Stress via Saliva in Surgical and Endoscopic Procedures and Its Measurement Intention in the Community: Reality-Future Prospects"

_children, 2023, doi:10.3390/children10050853_

Round 1

Reviewer 1 Report

Dear Authors,

I was very amazed by the idea of your research! I hope that your paper will be published.

Best regards,

Reviewer :)

Author Response

Dear Sir or Madam

Thank you for your comments, which are highly appreciated. We proceeded with the check of some spelling mistakes. Thank your for highlighting this problem to us.

Reviewer 2 Report

Dear authors, 

This is an interesting and important study  about how important is the positive attitude and correct information received by parents and children before medical and surgical interventions. I only have some minor comments.

-Line 108 and 126: was the lower age limit for the children 3 or 4?

- Line 169-170: "The greater cognitive capacity of older children before intervention S Cortisol level Τ2-Τ1 decreased by 5.21" - I do not understand this phrase. Please reformulate.

In the Discussion section please emphasize better the novelty (if there is any) or the importance of your study. 

Author Response

Dear Sir or Madam,

Thank you for your favourable comments. We really appreciate it. We corrected the problems which you highlighted. More specifically the lower age is 4 in line 130 (previously 128), we rephrased the sentence which was not clearly understood in lines 169-170 (previously 169-170)  and we have added the lines 280-293  at the end of the Discussion which we believe emphasizes the importance and novelty of our research. We also added some lines in our conclusion (280-293) which emphasize the importance of such methods for the future. We also checked for any mistakes concerning the English language and the spelling mistakes. We will also keep in mind your comments about the improvements regarding the methods, we added one sentence in the material and method for a better understanding (107-110) , the presentation of the results and the conclusions, which will be improved in the case that our research is accepted for publication in the journal.

Reviewer 3 Report

Interesting article. Without bringing revolutionary ideas it is a sound research.

Author Response

Dear Sir or Madam,

Thank you for your comments and your favourable review. We are very happy that you found our research interesting and sound. We really appreciate it.